

# An observational study on risk of secondary cancers in chronic myeloid leukemia patients in the TKI era in the United States

Vivek Kumar[1], Mohit Garg[2], Neha Chaudhary[3] and
Abhinav Binod Chandra[4]

[1] Department of General Internal Medicine, Brigham and Women's Hospital, Boston, MA, USA
[2] Department of Anesthesia, Maimonides Medical Center, New York, NY, USA
[3] Department of Pediatrics, Maimonides Medical Center, New York, NY, USA
[4] Department of Hematology and Oncology, Yuma Regional Medical Center Cancer Center,
Yuma, AZ, USA

## ABSTRACT

**Introduction:** The treatment with tyrosine kinase inhibitors (TKIs) has drastically improved the outcome of chronic myeloid leukemia (CML) patients. This study was conducted to examine the risk of secondary cancers (SCs) in the CML patients who were diagnosed and treated in the TKI era in the United States.

**Methods:** The surveillance epidemiology and end results (SEER) database was used to identify CML patients who were diagnosed and received treatment during January 2002–December 2014. Standardized incidence ratios (SIRs) and absolute excess risks (AER) were calculated.

**Results:** Overall, 511 SCs (excluding acute leukemia) developed in 9,200 CML patients followed for 38,433 person-years. The risk of developing SCs in the CML patients was 30% higher than the age, sex and race matched standard population (SIR 1.30, 95% CI: 1.2–1.40; $p < 0.001$). The SIRs for CLL (SIR 3.4, 95% CI: 2–5.5; $p < 0.001$), thyroid (SIR 2.2, 95% CI: 1.2–3.5; $p < 0.001$), small intestine (SIR 3.1, 95% CI: 1.1–7; $p = 0.004$), gingiva (SIR 3.7, 95% CI: 1.2–8.7; $p = 0.002$), stomach (SIR 2.1, 95% CI: 1.1–3.5; $p = 0.005$), lung (SIR 1.4, 95% CI: 1.1–1.7; $p = 0.006$) and prostate (SIR 1.3, 95% CI: 1.02–1.6; $p = 0.026$) cancer among CML patients were significantly higher than the general population. The risk of SCs was higher irrespective of age and it was highest in the period 2–12 months after the diagnosis of CML. The risk of SCs in women was similar to that of the general population.

**Conclusion:** CML patients diagnosed and treated in the TKI era in the United States are at an increased risk of developing a second malignancy. The increased risk of SCs in the early period after CML diagnosis suggests that the risk of SCs may be increased due to the factors other than TKIs treatment.

Corresponding author
Abhinav Binod Chandra,
abhinavbck@hotmail.com

## INTRODUCTION

The outcome of chronic myeloid leukemia (CML) patients is drastically changed by tyrosine kinase inhibitors (TKIs). The long-term survival of CML patients who have achieved complete cytogenetic remission is now similar to the general population (*Gambacorti-Passerini et al., 2011*). The increased survival of these patients require better understanding of long-term adverse effects of TKIs particularly development of de novo malignancies.

The carcinogenic potential of imatinib was first reported in a two-year carcinogenicity study on rats in which a dose depended risk due to imatinib was observed (*FDA, 2018*). The no observed effect level (NOEL) was 15 mg/kg/day. At dose of 30 mg/kg/day onwards (~0.5 or 0.3 times the human daily exposure at 400 or 800 mg/day, respectively), papilloma/carcinoma of the preputial/clitoral gland were noted. At dose of 60 mg/kg/day (~1.7 or 1 times the human daily exposure at 400 or 800 mg/day, respectively), the renal adenoma/carcinoma, the urinary bladder and urethral papilloma, the small intestine adenocarcinomas, the parathyroid glands adenomas, the benign and malignant medullary tumors of the adrenal glands and the non-glandular stomach papilloma/carcinomas were noted. However the relevance of these findings for humans are not yet clarified, despite many years of use of TKIs (*FDA, 2018*).

Many studies in the pre-TKI period reported increased risk of secondary cancers (SCs) in the CML patients as compared to the general population (*Frederiksen et al., 2011*; *Rebora et al., 2010*). The data on risk of SCs in the CML patients in the post TKI period are inconclusive. *Roy et al. (2005)* reported four times higher risk of prostate cancer in a study on 1,096 imatinib treated CML patients who were previously treated with interferon. In response to this, Pilot et al. reviewed Novartis registry of CML patients and concluded that the incidence of prostate cancer in imatinib treated patients was not higher than general population (SIR 0.87, 95% CI: 0.69–1.08) (*Roy et al., 2005*). Several other studies conducted in the post TKI era reported contrary data with increased risk (*Gunnarsson et al., 2015*), similar risk (*Gugliotta et al., 2017*; *Miranda et al., 2016*) or lower risk (*Verma et al., 2011*) than the general population. Also there was significant heterogeneity in the type of SCs reported in these studies. The higher incidence of gastrointestinal (GI), nose and throat, melanoma, kidney, endocrine and non-Hodgkin's lymphoma (NHL) has been reported (*Gunnarsson et al., 2015*; *Miranda et al., 2016*; *Verma et al., 2011*). Previous studies based on SEER database analyzed data in the pre-TKI era (till 2002) and reported 16% higher risk of SCs in CML survivors while another study compared pre and post TKI treatment risks and reported approximately 50% higher risk of developing SCs in CML patients diagnosed during 2002–2009 as compared to the general population in the United States (*Brenner, Gondos & Pulte, 2009*; *Shah & Ghimire, 2014*). However, the actual treatment status of these patients was not disclosed in SEER database at that time. The SEER database has released customized database in April 2017 with information on chemotherapy (*National Cancer Institute, 2017b*). However, it did not specify the type of treatment individual patient received. Although it can be safely assumed that most of the patients with CML who were diagnosed in the post TKI era, received TKIs

as they are the treatment of choice for CML since FDA approval in May 2001. Currently, other therapies are restricted to a small number of patients who are resistant to second generation TKIs or harbor TKIs resistant mutations like T3151 (*Jain & van Besien, 2011*).

This study aimed to analyze the risk of SCs among patients with CML in the TKI era in the U.S.

## MATERIALS AND METHODS

### The SEER database

The SEER program is a population based registry which is maintained by the National Cancer Institute and covers approximately 28% of the U.S. population (*National Cancer Institute, 2017a*). It publishes data on patient demographics, cancer trends, SCs, outcome, and follow-up. We analyzed data from the SEER-18 (2000–2014) database, released in April 2017.

### Study population

Patients >20 years old who were diagnosed of CML between January 2002 and December 2014, were identified using SEER*Stat, version 8.3.4 multiple primary-standardized incidence ratio (MP-SIR) session. The customized dataset contains information on chemotherapy as "yes" or "no/unknown" variables. The chemotherapy has been charted as "yes" in the SEER if patient records confirmed treatment. As per the SEER* Rx Interactive Antineoplastic Drug Database, chemotherapy has been recorded "yes" for the patients who received any of the TKIs including imatinib, ponatinib, bosutinib, dasatinib and nilotinib or any other conventional chemotherapy. For this study we only included patients who were coded as "yes" for receiving treatment. The imatinib was approved for the treatment of CML in the US by FDA in May 2001 (*Cancer Network, 2001*). TKIs (imatinib or its congeners) are the drug of choice for CML patients since their FDA approval and most of the CML patients in the US have been treated with imatinib or one of the newer TKIs. We collected data on demographics, date of diagnosis of CML and SCs excluding non-melanoma skin cancers, type of SCs, vital status, cause of death, and overall survival.

### Patient selection

A query was run in SEER stat software to identify all the patients of age 20 years old and above who were diagnosed of CML and were coded "**yes**" for the chemotherapy between January 2002 and December 2014. Patients were excluded if they survived for less than two months (to adjust for surveillance bias) from the date of CML diagnosis, if CML was diagnosed on autopsy or if CML was not the primary cancer. A total of 9,341 patients were identified who met the eligibility criteria. Patients were also excluded if they received external beam radiation therapy as the initial treatment because most likely these patients underwent hematopoietic stem cell transplantation (HSCT) as described in the discussion below. A total of 141 (<2%) patients who developed six SCs were excluded. Finally, 9,200 patients were included in the analysis. The inherent risk of development of AML and ALL (blast transformation) is well known in the CML patients. The cases of acute leukemia were not included in the estimation of overall SCs risk. A sensitivity analysis was

conducted after excluding SCs which were diagnosed during the first year after diagnosis of CML to adjust for the surveillance bias.

## Statistical analysis

The risk of SCs in CML patients was evaluated by accumulating person-years (PYs), sex, and calendar-year from two months after diagnosis of CML to the date of death, last follow-up, diagnosis of SC, or the study end (December 31, 2014), whichever occurred first. Expected SCs in the CML population were calculated based on the 2,000 U.S. standard population distribution, by multiplying the incidence rates specific for sex, race, five-year attained age, and calendar-year by the specific PYs at risk, followed by its summation as incorporated in SEER*stat, version 8.3.4. Standardized incidence ratios (SIRs) were expressed as the ratio of observed to expected events. The absolute excess risk (AER per 10,000 PYs) was estimated by subtracting the expected from the observed number of SCs and dividing the difference by the number of PYs at risk.

A Poisson distribution of observed SCs was assumed for calculation of the 95% confidence intervals (CIs) and "$p$" value.

## RESULTS

A total of 9,200 patients were eligible for the study. These patients were followed for an average of 4.2 years accumulating 38,433 PYs. The demographic characteristics of study patients are shown in Table 1. Briefly, 41% were females, 80% were white and 44% were of age above 60 years.

Overall, 511 SCs were diagnosed during the study period. The distribution of selected SCs (where at least five cancers were diagnosed) with their SIRs and excess risks have been shown in Fig. 1. The risk of developing SCs in the CML patients was 30% higher than the age, sex and race matched standard population (SIR 1.3, 95% CI: 1.2–1.4; $p < 0.001$). This aggregated to an excess of 30 cancers per 10,000 PYs. The absolute risk of developing a SCs was 1.3% per year (511/38,433) in the survivors of CML.

Of 511 cancers, 94 (18%) were localized to the GI tract, 90 (18%) were in the prostate, 77 (15%) were lung cancer and 78 (15%) were hematological malignancies (excluding AML and ALL). The SCs whose risks were more than three times of general population included gingiva (SIR 3.7, 95% CI: 1.2–8.7; $p = 0.002$), CLL (SIR 3.4, 95% CI: 2–5.5; $p < 0.001$) and small intestine (SIR 3.1, 95% CI: 1.2–7; $p = 0.004$). The risk of thyroid (SIR 2.2, 95% CI: 1.2–3.5; $p < 0.001$) and stomach (SIR 2.1, 95% CI: 1.1–3.5; $p = 0.005$) cancers was doubled in the survivors of CML. The risk for developing melanoma (SIR 1.5, 95% CI: 1.1–2.2; $p = 0.024$), lung cancer (SIR 1.4, 95% CI: 1.1–1.7; $p = 0.006$) and prostate cancer (SIR 1.3, 95% CI: 1.02–1.6; $p = 0.026$) was also significantly higher than the general population (Fig. 1).

The increased risk of SC was observed only in the men who were at 40% (SIR 1.4, 95% CI: 1.3–1.7; $p < 0.001$) higher risk of developing SCs after the diagnosis of CML. This contributed to 43 excess cancers in men per 10,000 PYs (Table 2). On the other hand, in women, the risk of SC was similar to the general population (SIR 1.1, 95% CI: 0.9–1.3; $p = 0.11$). Also, the individual cancer risk was not different in the women compared to

Table 1 Demographic characteristics of study population.

| Demographic characteristics | N = 9,200 (100%) |
|---|---|
| *Gender* | |
| Male | 5,420 (59) |
| Female | 3,780 (41) |
| *Age (in years)* | |
| <60 | 5,190 (56) |
| ≥60 | 4,010 (44) |
| *Ethnicity* | |
| White | 7,338 (80) |
| African-American | 1,050 (11) |
| American Indian/Alaska Native | 55 (<1) |
| Asian/Pacific Islander | 625 (7) |
| Unknown | 132 (1) |
| *Marital status* | |
| Married | 5,067 (55) |
| Single | 1,756 (19) |
| Previously married | 1,654 (18) |
| Unknown | 723 (8) |
| *Geographical location* | |
| Northern plains | 994 (11) |
| East | 3,568 (39) |
| Pacific coast | 4,222 (46) |
| Southwest | 416 (4) |
| *Outcome at study cut-off* | |
| Alive | 6,397 (70) |
| Dead | 2,803 (30) |

the general population, with the exception of gastric cancer whose risk was three-times higher (SIR 3.5, 95% CI: 1.4–7.3; $p < 0.001$), colon cancer (SIR 1.7, 95% CI: 1.03–2.7; $p = 0.02$) and breast cancer whose risk was lower than the general population (SIR 0.6, 95% CI: 0.4–0.9; $p = 0.009$) (Table 2).

When assessed by age at diagnosis of CML, 190 (37%) SCs were diagnosed in the patients under age of 60 years while 321 (63%) SCs were diagnosed in the patients above 60 years of age. The risk of developing SCs was 50% higher in the patients below 60 years of age and 20% higher in patients above 60 years of age compared to the general population. Patients below 60 years developed more CLL, skin melanoma and thyroid cancers compared to the general population while elderly patients were at significantly higher risk of developing cancers of gingiva, soft tissues including heart and lungs (Figs. 2 and 3).

## Follow-up of study population

The highest risk of SCs was observed in the period 2–11 months after the diagnosis of CML (SIR 1.4, 95% CI: 1.1–1.8; $p < 0.001$) (Table S1). The risk of developing any SC remained elevated up to five years from the diagnosis of CML. However, approximately

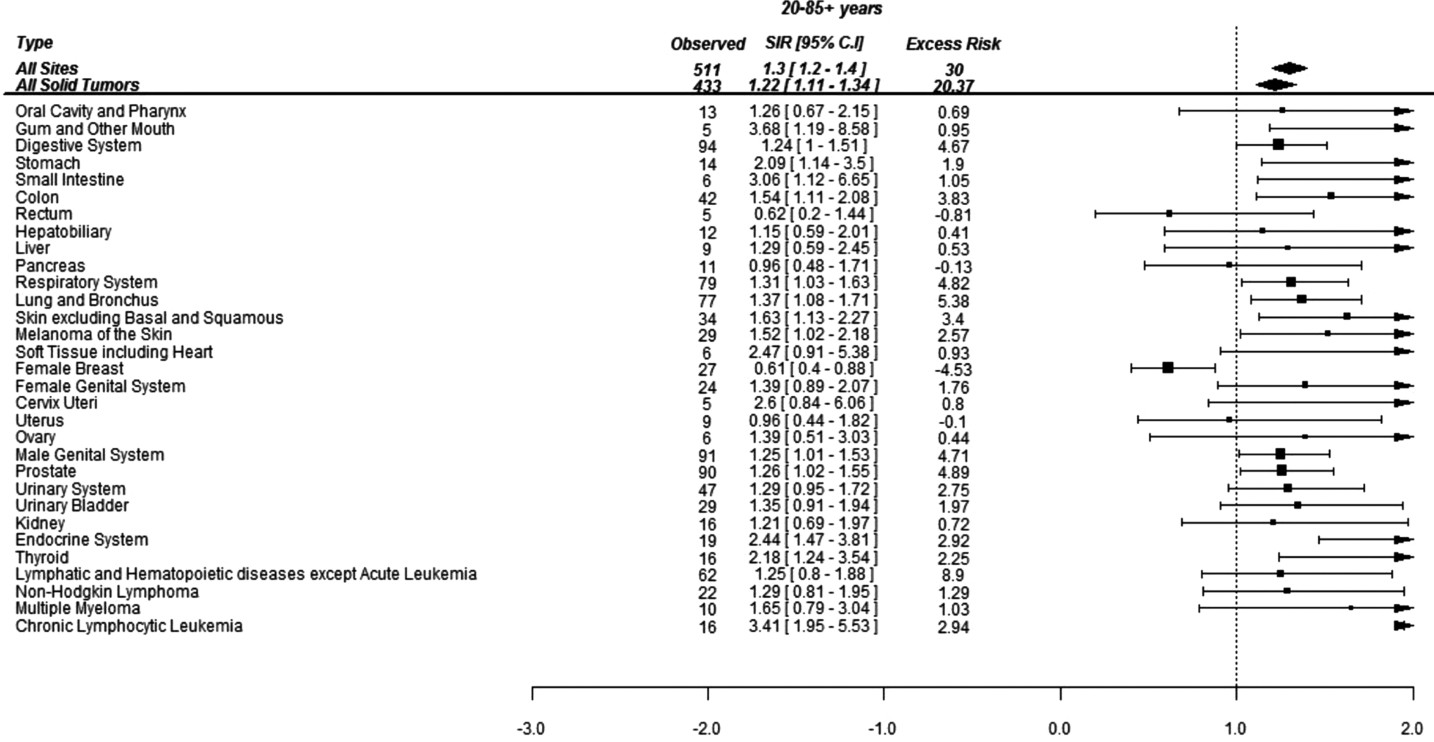

**Figure 1** Standardized incidence ratios (SIR) and absolute excess risk (AER) of selected secondary cancers in CML patients. Absolute excess risk is per 10,000 individuals.

only one-third patients were followed for more than five years (3,395 patients for 9,809 PYs). The higher SIRs of CLL, colon and endocrine cancers were apparent within the first year after the diagnosis of CML and remained elevated in the period one to five years after the diagnosis of CML. Other cancers whose risks were higher during one to five years included prostate, cervix and lung and bronchus. After five years, although the overall SIR of SCs was similar to the U.S. general population, the individual risks of tumors of gingiva, skin and stomach were higher. After 10 years the overall risk of only hematological cancers was high due to the higher risk of NHL (Table S1).

A sensitivity analysis was conducted by excluding all patients who survived for less than one year after diagnosis of CML to adjust for the surveillance bias. Though many cases were excluded (106 SCs were excluded), the overall risk of SCs did not change (Table 3).

## DISCUSSION

This large population based longitudinal analysis revealed that the CML patients who were diagnosed and treated in the TKI era were at 30% higher risk of developing SCs as compared to the general population. The higher risk was seen in younger as well as elderly patients but was limited only to men. The risk was higher for multiple cancers including CLL, small intestine, gingiva, thyroid, melanoma, lung and prostate cancer. The overall risk of developing SCs was higher for up to five years after the diagnosis of CML.

**Table 2 SIR and Excess risk of Secondary Cancers among patients with CML stratified by gender.**

| Cancer sites | Male | | | | Female | | | |
|---|---|---|---|---|---|---|---|---|
| | Observed | SIR (95% CI) | Excess risk | 'p' Value | Observed | SIR (95% CI) | Excess risk | 'p' Value |
| All sites* | 342 | 1.4 (1.25–1.65) | 43.13 | **<0.001** | 169 | 1.14 (0.94–1.34) | 9.85 | 0.33 |
| All solid tumors | 285 | 1.29 (1.15–1.45) | 28.97 | **<0.001** | 148 | 1.1 (0.93–1.3) | 8.61 | 0.24 |
| Oral cavity and pharynx | 11 | 1.37 (0.68–2.44) | 1.33 | 0.29 | 2 | 0.87 (0.11–3.14) | −0.19 | 0.84 |
| Gum and other mouth | 4 | 4.66 (1.3–11.93) | 1.41 | **<0.001** | 1 | 1.99 (0.05–11.11) | 0.31 | 0.49 |
| Digestive system | 58 | 1.19 (0.9–1.53) | 4.08 | 0.18 | 36 | 1.33 (0.93–1.84) | 5.46 | 0.09 |
| Stomach | 7 | 1.48 (0.6–3.05) | 1.02 | 0.29 | 7 | 3.53 (1.42–7.27) | 3.09 | **<0.001** |
| Small intestine | 3 | 2.43 (0.5–7.09) | 0.79 | 0.11 | 3 | 4.13 (0.85–12.06) | 1.4 | 0.06 |
| Colon | 23 | 1.42 (0.9–2.14) | 3.08 | 0.09 | 19 | 1.71 (1.03–2.67) | 4.87 | **0.02** |
| Rectum | 5 | 0.91 (0.3–2.13) | −0.21 | 0.83 | 0 | 0 (0–1.4) | −1.63 | 0.8 |
| Hepatobiliary system | 10 | 1.32 (0.63–2.43) | 1.1 | 0.38 | 2 | 0.69 (0.08–2.51) | −0.54 | 0.6 |
| Liver | 7 | 1.25 (0.5–2.57) | 0.63 | 0.55 | 2 | 1.47 (0.18–5.3) | 0.39 | 0.58 |
| Pancreas | 6 | 0.87 (0.32–1.89) | −0.41 | 0.73 | 5 | 1.09 (0.35–2.54) | 0.25 | 0.85 |
| Respiratory system | 48 | 1.24 (0.91–1.64) | 4.17 | 0.13 | 31 | 1.43 (0.97–2.03) | 5.72 | 0.06 |
| Lung and bronchus | 48 | 1.36 (1–1.8) | 5.72 | **0.03** | 29 | 1.38 (0.92–1.98) | 4.91 | 0.08 |
| Soft tissue and heart | 5 | 3.14 (1.02–7.33) | 1.53 | **0.007** | 1 | 1.2 (0.03–6.66) | 0.1 | 0.85 |
| Skin (except basal/squamous) | 25 | 1.68 (1.09–2.48) | 4.55 | **0.008** | 9 | 1.5 (0.68–2.84) | 1.84 | 0.22 |
| Melanoma (skin) | 21 | 1.54 (0.95–2.36) | 3.33 | 0.06 | 8 | 1.46 (0.63–2.87) | 1.55 | 0.28 |
| Female breast | NA | NA | NA | – | 27 | 0.61 (0.4–0.88) | −10.73 | **0.009** |
| Female genital system | NA | NA | NA | – | 24 | 1.39 (0.89–2.07) | 4.17 | 0.1 |
| Cervix uteri | NA | NA | NA | – | 5 | 2.6 (0.84–6.06) | 1.89 | 0.08 |
| Corpus and uterus, NOS | NA | NA | NA | – | 9 | 0.96 (0.44–1.82) | −0.25 | 0.89 |
| Ovary | NA | NA | NA | – | 6 | 1.39 (0.51–3.03) | 1.04 | 0.42 |
| Male genital system | 91 | 1.25 (1.01–1.53) | 8.16 | **0.03** | NA | NA | NA | – |
| Prostate | 90 | 1.26 (1.02–1.55) | 8.46 | **0.02** | NA | NA | NA | – |
| Urinary system | 38 | 1.33 (0.94–1.83) | 4.26 | **0.08** | 9 | 1.14 (0.52–2.17) | 0.69 | 0.69 |
| Urinary bladder | 26 | 1.47 (0.96–2.15) | 3.72 | 0.06 | 3 | 0.81 (0.17–2.37) | −0.43 | 0.72 |
| Kidney | 11 | 1.15 (0.57–2.06) | 0.65 | 0.64 | 5 | 1.37 (0.44–3.19) | 0.83 | 0.48 |
| Endocrine system | 10 | 3.45 (1.66–6.35) | 3.2 | **<0.001** | 9 | 1.84 (0.84–3.49) | 2.53 | 0.06 |
| Thyroid | 8 | 3.04 (1.31–5.99) | 2.42 | **<0.001** | 8 | 1.7 (0.73–3.34) | 2.02 | 0.13 |
| Hematological system* | 47 | 2.2 (1.1–3.3) | 10.82 | **<0.001** | 19 | 1.7 (0.5–3) | 1.2 | 0.1 |
| Non-Hodgkin lymphoma | 15 | 1.38 (0.77–2.27) | 1.85 | 0.22 | 7 | 1.14 (0.46–2.34) | 0.51 | 0.73 |
| Myeloma | 7 | 1.77 (0.71–3.64) | 1.37 | 0.13 | 3 | 1.44 (0.3–4.21) | 0.57 | 0.53 |
| Chronic lymphocytic leukemia | 13 | 3.99 (2.13–6.82) | 4.39 | **<0.001** | 3 | 2.08 (0.43–6.09) | 0.96 | 0.2 |

**Notes:**
NA, not applicable. Numbers in bold indicate that the p value is significant at $p < 0.05$.
* After excluding acute leukemia.

A previous study by Shah et al. involving CML patients diagnosed in SEER database during 1992–2009 compared the risk of SCs using year of diagnosis as proxy of treatment without actual treatment data (*Shah & Ghimire, 2014*). They concluded that the risk of SCs increased in the cohort which were diagnosed and treated in the study period
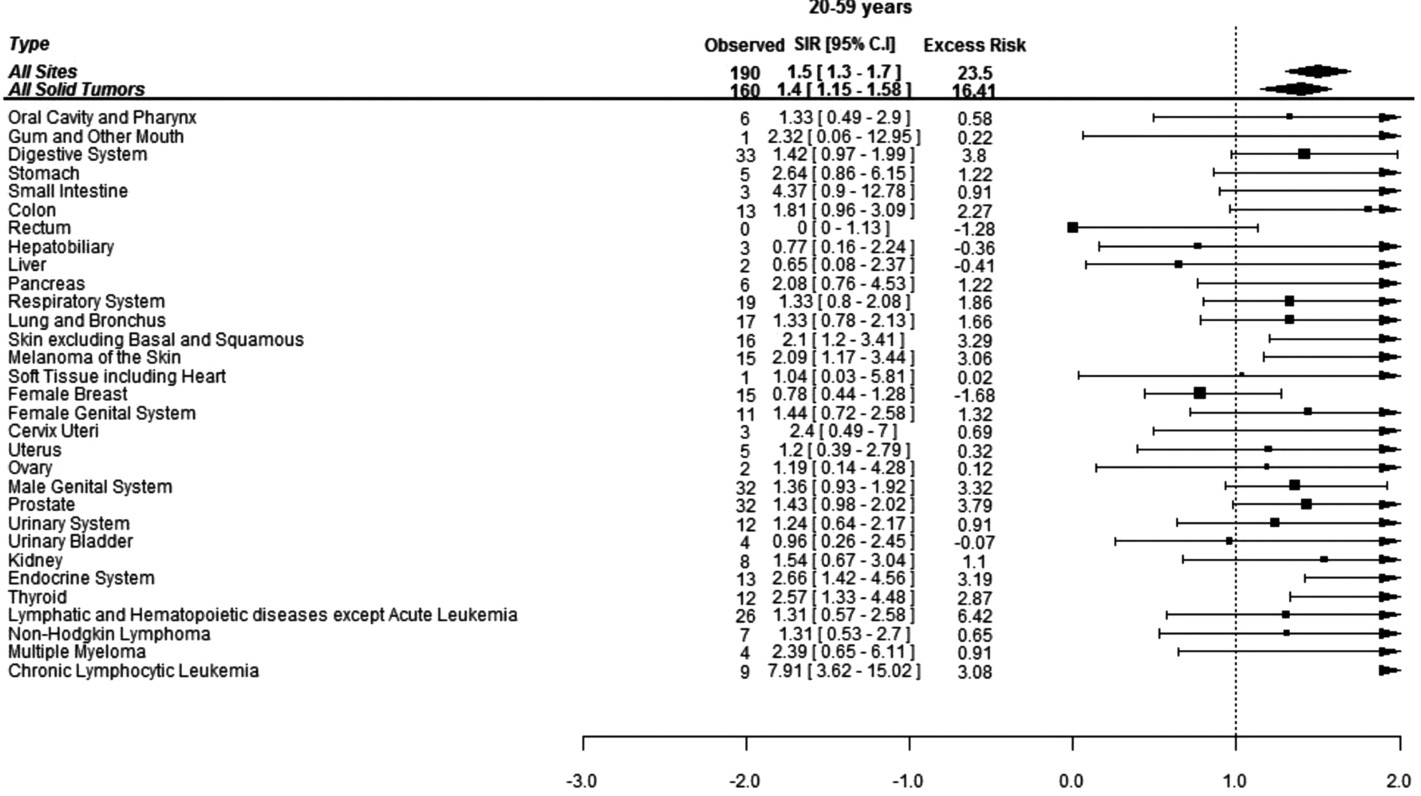

**Figure 2 Standardized incidence ratios (SIR) and absolute excess risk (AER) of selected secondary cancers in CML patients aged 20–59 years.** Absolute excess risk is per 10,000 individuals.

corresponding to TKI era. The risk in the cohort diagnosed during 2002–2009 was estimated to be 49% higher than the general population. The present study extends the data to 2014 and included only patients who were diagnosed and received treatment in the TKI period. The lower (30%) risk in this study is likely due to the exclusion of acute leukemia from the analysis similar to previous studies (*Gugliotta et al., 2017*; *Gunnarsson et al., 2015*).

The higher risk observed in our study has also been reported previously. *Voglova et al. (2011)* reported 1.5 times higher risk of developing SCs among 1,038 patients with CML treated during 2000–2009. The patients were followed for mean duration of 58 months (2–214 months) after starting TKIs. However in that study risks of SCs at individual sites were similar to the general population. The results from this study are also concordant with the results from another large population study based upon Swedish CML-register by Gunnarson et al. They reported that CML patients diagnosed during 2002–2011, were at 50% higher risk (SIR 1.5, 95% CI: 1.3–2) of developing SCs as compared to the general population. After a follow-up of 3.7 years, 7.5% patients developed SCs (*Gunnarsson et al., 2015*). They reported significantly higher SIRs in older patients and for cancers at certain sites like GI, nose and throat. However, contrary to the present study, the risk was higher among women as compared to men. Besides, higher risk

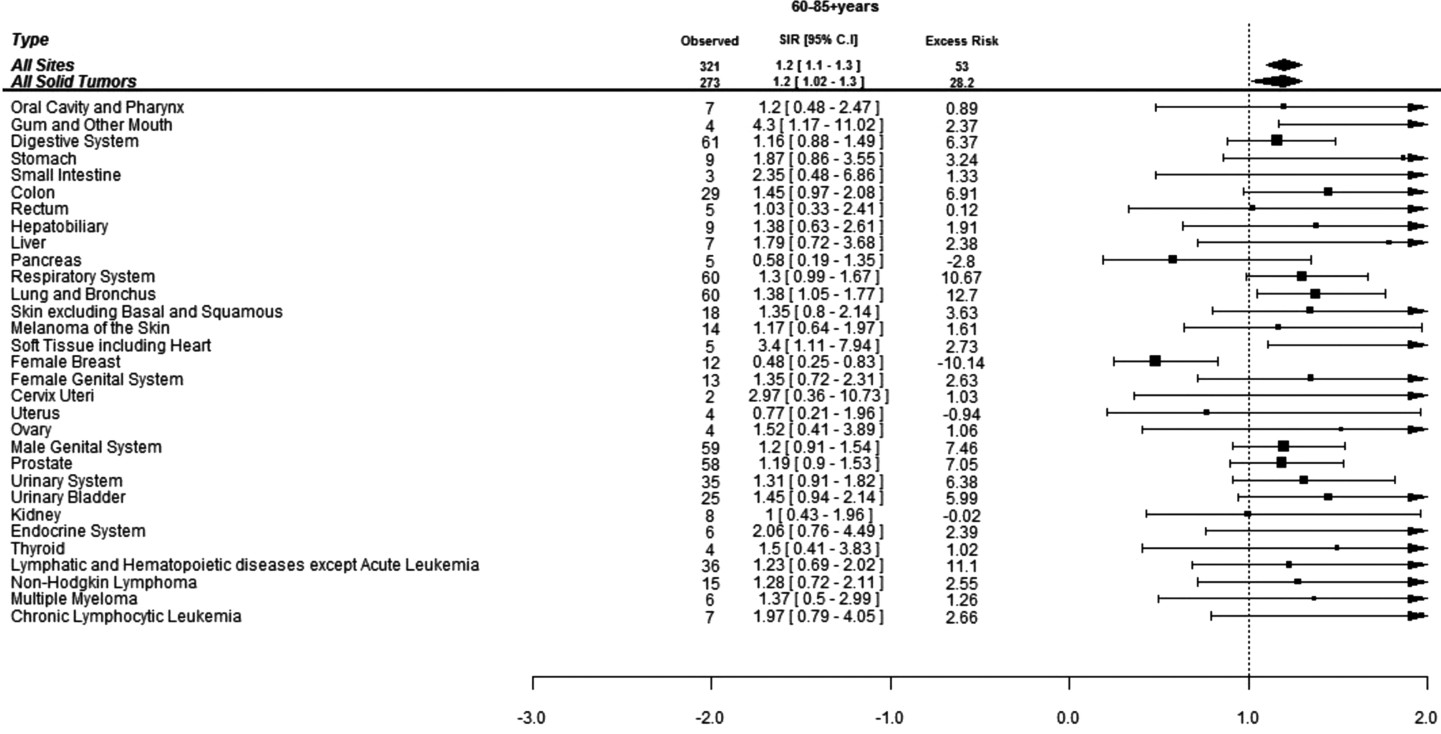

**Figure 3 Standardized incidence ratios (SIR) and absolute excess risk (AER) of selected secondary cancers in CML patients aged 60–85+ years.** Excess risk is per 10,000 individuals.

of SCs in CML patients has also been reported in several small studies (*Duman et al., 2012*; *Helbig et al., 2015*; *Pilot et al., 2006*).

On contrary the findings in our study are discordant to that reported by *Miranda et al. (2016)*. They analyzed data from the CML IV study on 1,525 patients who were followed for a median of 67.5 months. The overall risk of SCs (SIR 0.9, 95% CI: 0.6–1.2) was not higher than the general population but they reported significantly higher risk of NHL among men and women. In another study by *Verma et al. (2011)*, 103 SCs were diagnosed in 1,342 CML patients (median follow-up of 107 months) treated with TKIs during 1998–2010. Overall significantly fewer cancers were reported than expected for the cohort with SIR of 0.6 (95% CI: 0.4–0.8). However, the number of SCs in this study could have been underestimated since only the first cancer was reported. Nonetheless, the risk of certain types of SCs like melanoma, kidney and endocrine system was higher than the age, sex and race matched standard population. The risk was lower than expected for tumors at breast, prostate and digestive tract. Similarly, *Gugliotta et al. (2017)* reported no significant increase in the risk of SCs among CML patients enrolled in imatinib trials.

The higher risk of certain cancers such as CLL, small intestine, gingiva, thyroid, melanoma, lung and prostate cancer as compared to general population were noted in the present study. There is no consistency in the pattern of SCs reported in various studies (*Gunnarsson et al., 2015*; *Miranda et al., 2016*; *Pilot et al., 2006*; *Verma et al., 2011*).

**Table 3  SIRs and excess risks after excluding secondary cancers which were diagnosed within first year after the diagnosis of CML.**

| Cancer sites | Observed | O/E | CI lower | CI upper | Excess risk | 'p' Value |
|---|---|---|---|---|---|---|
| All sites excluding non-melanoma skin* | 405 | 1.26 | 1.12 | 1.4 | 26.45 | **<0.001** |
| All solid tumors | 350 | 1.21 | 1.09 | 1.34 | 19.27 | **<0.001** |
| Oral cavity and pharynx | 12 | 1.41 | 0.73 | 2.46 | 1.11 | 0.23 |
| Gum and other mouth | 4 | 3.6 | 1.3 | 9.22 | 0.92 | **0.006** |
| Digestive system | 76 | 1.23 | 0.97 | 1.54 | 4.46 | 0.08 |
| Stomach | 12 | 2.20 | 1.14 | 3.85 | 2.08 | **0.004** |
| Small intestine | 4 | 2.47 | 0.67 | 6.33 | 0.76 | 0.06 |
| Colon | 31 | 1.41 | 0.96 | 2 | 2.85 | 0.83 |
| Rectum | 4 | 0.61 | 0.16 | 1.55 | −0.83 | 0.32 |
| Hepatobiliary system | 11 | 1.28 | 0.64 | 2.28 | 0.76 | 0.42 |
| Liver | 9 | 1.56 | 0.71 | 2.96 | 1.02 | 0.18 |
| Pancreas | 10 | 1.06 | 0.51 | 1.95 | 0.18 | 0.85 |
| Respiratory system | 62 | 1.26 | 0.97 | 1.62 | 4.1 | 0.07 |
| Lung and bronchus | 60 | 1.31 | 1 | 1.69 | 4.53 | **0.03** |
| Soft tissue including heart | 6 | 3.01 | 1.11 | 6.56 | 1.27 | **0.004** |
| Skin excluding basal and squamous | 26 | 1.51 | 0.99 | 2.22 | 2.79 | 0.06 |
| Melanoma of the skin | 22 | 1.4 | 0.88 | 2.12 | 1.99 | 0.11 |
| Female breast | 23 | 0.63 | 0.4 | 0.94 | −4.35 | **0.03** |
| Female genital system | 22 | 1.54 | 0.97 | 2.33 | 2.45 | 0.07 |
| Cervix uteri | 4 | 2.53 | 0.69 | 6.47 | 0.77 | 0.06 |
| Corpus uteri | 9 | 1.19 | 0.54 | 2.25 | 0.45 | 0.67 |
| Ovary | 5 | 1.41 | 0.46 | 3.29 | 0.46 | 0.44 |
| Male genital system | 76 | 1.29 | 1.02 | 1.62 | 5.48 | **0.03** |
| Prostate | 75 | 1.31 | 1.03 | 1.64 | 5.59 | **0.02** |
| Urinary system | 37 | 1.25 | 0.88 | 1.72 | 2.33 | 0.17 |
| Urinary bladder | 21 | 1.21 | 0.75 | 1.85 | 1.16 | 0.38 |
| Kidney | 14 | 1.29 | 0.7 | 2.16 | 0.99 | 0.35 |
| Endocrine system | 12 | 1.85 | 0.95 | 3.23 | 1.74 | 0.12 |
| Thyroid | 12 | 1.95 | 1.01 | 3.41 | 1.86 | **0.02** |
| Hematological malignancies* | 47 | 1.8 | 1.1 | 2.5 | 6.68 | **<0.001** |
| Non-Hodgkin lymphoma | 17 | 1.22 | 0.71 | 1.95 | 0.97 | 0.42 |
| Myeloma | 8 | 1.62 | 0.7 | 3.18 | 0.97 | 0.16 |
| Chronic lymphocytic leukemia | 11 | 2.87 | 1.43 | 5.14 | 2.27 | **<0.001** |

Notes:
Excess risk is per 10,000. Confidence intervals are 95%. Numbers in bold indicate that the $p$ value is significant at $p < 0.05$.
* Acute leukemia excluded.

Higher risks of GI, hepatobiliary, adrenal and hematological malignancies were reported in the previous study based on SEER database (*Shah & Ghimire, 2014*). The authors reported no change in the risk of individual cancers in the pre and post TKI periods. However in the current analysis higher risks were also noted for gingiva, thyroid, lung and prostate cancers. Cancers at gingiva, thyroid and small intestine contributed little to excess risk despite high SIRs due to their low rate in the background population. The etiology of

increased risk at selected sites in the CML patients is unclear and could be due to the higher prevalence of risk factors like tobacco in these patients. Unfortunately, there is no data on risk factors in the SEER database to confirm this hypothesis. Moreover the higher SIRs for gingiva and thyroid cancers were based on a very small number of cases which makes its interpretation difficult. The risk of hematological cancers were not higher than the general population after excluding acute leukemia but the individual risk of CLL was higher in the first year after diagnosis of CML while the risk of NHL was higher after 10 years from the diagnosis of CML. Nonetheless, very few patients were followed beyond 10 years, more data is required to ascertain the higher risk of NHL in the CML patients. The lower risk of breast cancer in the women with CML similar to our study has also been reported by *Verma et al. (2011)* (SIR 0.24, 0.03–0.9). The reason for this interesting finding is not clear but the gonadotoxicity (and resulting ovarian failure which could be protective against breast cancer) due to imatinib has not been established yet. Although few case reports and preliminary data suggested premature ovarian failure among the patients with CML this has not been specifically tested in large prospective studies (*Christopoulos, Dimakopoulou & Rotas, 2008*).

Other interesting finding from the current study was the rare occurrence of CLL among patients with CML. The coexistence of CLL and CML has been described in the literature in anecdotal reports (*D'Arena et al., 2012*; *Gargallo et al., 2005*). In a case where CLL followed CML six years after its diagnosis, the genomic studies suggested separate origins for myeloid and lymphoid clones which carried mutually exclusive positive genomic markers (del17q11 (CLL) and *BCR/ABL (CML)*), supporting the two genomic events/two diseases hypothesis (*D'Arena et al., 2012*). In that patient, CLL responded to a second generation TKI, dasatinib which is also a treatment for CML. In the present study, CLL was diagnosed among 0.17% patients. Six of these patients were diagnosed within the first year of diagnosis of CML and possibility of pre-existing CLL can't be ruled out in these patients. In contrast to the case reported previously, majority of these patients were males with approximately four times higher risk of developing CLL as compared to the U.S. general population. The association of CLL with CML and male predisposition warrant further studies.

The reason for higher risk of SCs in CML is not clear. Imatinib has immunosuppressive properties by virtue of its inhibitory effect on differentiation of dendritic cells (DCs) from CD 34+ progenitor cells (*Rea et al., 2004*). The resulting cells in the presence of imatinib although bear resemblance to normal DCs but have lower expression of surface molecules like CD1a, CD 38, major histocompatibility complex (MHC) II, thus these cells are unable to mount T-cell response. Rea et al. reported that in patients on imatinib treatment, DCs differentiation rates and Th1/Th2 balance remained impaired despite normalization of vascular endothelial growth factors (VEGF). Imatinib also inhibits T-cells proliferation by arresting cells in G0/G1 phase (*Rea et al., 2004*). This may be more relevant to the newer TKIs with higher immunosuppressive effects (*Appel et al., 2005*). Other possible mechanism include interference by imatinib with the DNA repair mechanisms (*Majsterek et al., 2006*). However the carcinogenicity of TKIs has not been proven clearly in the clinical trials.

The SCs arising as a result of therapy are not expected to manifest until after several years of treatment as suggested in several Hodgkin's lymphoma studies where relative risk for SCs was the highest 5–10 years after the diagnosis of lymphoma (*Schaapveld et al., 2015*). In our study, the maximum risk of SCs, was seen in the period soon after CML diagnosis. The risk persisted in the first five years from the diagnosis of CML but only fewer patients were followed past five years and long-term follow-up studies are required to establish the period of risk. The more likely explanation for increased SCs in the period soon after the diagnosis includes factors other than TKIs like increase surveillance for other cancers or genetic predisposition due to CML itself (*Stein, 2012*). Unfortunately, our study was not designed to establish the causation of SCs and this hypothesis requires confirmation through the clinical trials or analysis of individual-level data from exclusive CML registries. Nonetheless, the increased risk in these patients mandates long-term active surveillance for the SCs.

Other possible etiologies for increased SCs in CML may include several disease related factors. BCR/ABL regulates apoptosis, proliferation and intercellular interactions. It also amplifies DNA damage and promote genomic instability which may increase the genetic susceptibility to acquire cancers other than CML (*Pawlowska & Blasiak, 2015*; *Skorski, 2008*). The studies on population based registries have reported higher risk of developing SCs contrary to the randomized trials. It has been hypothesized that due to better disease control in randomized trials, the propensity to develop SCs remain suppressed (*Miranda et al., 2016*). However, this has not been confirmed yet.

Chemotherapies which were used prior to the introduction of TKIs like busulphan have shown to be carcinogenic (*Majhail et al., 2011*). Moreover patients who have undergone HSCT are also at higher risk of developing SCs (*Tanaka et al., 2015*). Total body irradiation (TBI) in combination with cyclophosphamide was the preferred regimen for conditioning prior to transplant in the past (*Jain & van Besien, 2011*). In the more recent times non-myeloablative regimens or reduced intensity conditioning (RIC) are preferred and TBI (at 200 cGY) with fludarabine is one of them. The rate of HSCT has fallen drastically in the post TKI era. In an analysis from Europe, HSCT rate dropped by 69% in the year 2007 as compared to 1999 (*Gratwohl et al., 2006*). Currently HSCT is reserved for patients with CML after failure of second generation TKIs and among patients with TKI resistant mutations like T3151 (*Jain & van Besien, 2011*). The studies from pre-TKI era reported that SCs were the cause of deaths in up to 7% patients with CML after 10 years of follow-up (*Goldman et al., 2010*). There was no information on patients undergoing HSCT in the SEER database. However, patients who received radiation as part of the initial treatment of CML most likely underwent HSCT. Besides, radiotherapy is independently associated with increased risk of SCs (*Bartkowiak et al., 2012*). In this study a small number of patients who received RT were thus excluded from the analysis because it was difficult to dissect the impact of radiation treatment from TKIs on the SCs risks. Ideally, these patients should have been censored but this was not possible due to the limitation of current dataset.

The strengths of this study include its large sample size. This study included patients who actually received treatment in the TKI era. The information on treatment in SEER is available as "yes" or "no/unknown," and is 68% complete as compared to SEER-Medicare data. However, the specificity and positive predictive values are high which means that if chemotherapy is documented "yes," patient had most likely received it (*Noone et al., 2016*). Thus, this data supports the analysis on adverse events like SCs. Moreover the study is based on a population-based registry and more accurately reflects the risk in the community, outside the controlled settings.

However, the findings of this study should be interpreted with caution due to the following limitations. The information on treatment in the SEER database was available as "yes" or "no/unknown." There was no information on the type of treatment or patient adherence. However, it could be safely assumed that most of these patients diagnosed in the TKI era received one of the TKIs. Due to the large sample size, the small number of patients who could have received alternative treatment would have little effect on overall analysis. The data on HSCT was also not available in the SEER database. We excluded small number of patients who received radiation treatment most likely as part of conditioning agent prior to HSCT, which could have affected the development of SCs. Studies have reported disparity in the reporting of TKI treatment among the elderly patients in the population-based registries which could have misclassified some elderly patients into the non-treated group which were not included in the analysis (*Hoglund et al., 2013*; *Styles et al., 2016*). Lastly, the data on the cancer risk factors like smoking and genetic predisposition was not available in the SEER database.

In conclusion, the risk of developing SCs in CML patients in the US who were diagnosed and treated after the approval of TKIs was significantly higher than the general population. Though the cause of elevated risk is not clear, the diagnosis of SCs in the early period after CML diagnosis suggests the linkage to CML itself rather than TKIs. Further studies are warranted for its confirmation.

## ACKNOWLEDGEMENTS

This study used the SEER database. The interpretation and reporting of these data are the sole responsibility of the authors. The authors acknowledge the efforts of the National Cancer Institute; the Office of Research, Development and Information, CMS; Information Management Services (IMS), Inc.; and the Surveillance, Epidemiology, and End Results (SEER) Program tumor registries in the creation of the SEER database.

### Funding

This project was partly funded by the Foundation of Yuma Regional Medical Center on behalf of the Richard Michael McDaniel Endowment Fund for Cancer Care. The funders had no role in study design, data collection and analysis, decision to publish, or preparation of the manuscript.

## Grant Disclosures

The following grant information was disclosed by the authors:
Foundation of Yuma Regional Medical Center on behalf of the Richard Michael McDaniel Endowment Fund for Cancer Care.

## Competing Interests

The authors declare that they have no competing interests.

## Author Contributions

- Vivek Kumar conceived and designed the experiments, performed the experiments, analyzed the data, contributed reagents/materials/analysis tools, wrote the paper, prepared figures and/or tables, reviewed drafts of the paper.
- Mohit Garg conceived and designed the experiments, performed the experiments, analyzed the data, contributed reagents/materials/analysis tools, wrote the paper, prepared figures and/or tables, reviewed drafts of the paper.
- Neha Chaudhary conceived and designed the experiments, performed the experiments, analyzed the data, contributed reagents/materials/analysis tools, wrote the paper, prepared figures and/or tables, reviewed drafts of the paper.
- Abhinav Binod Chandra conceived and designed the experiments, performed the experiments, analyzed the data, contributed reagents/materials/analysis tools, wrote the paper, prepared figures and/or tables, reviewed drafts of the paper.

## Human Ethics

The following information was supplied relating to ethical approvals (i.e., approving body and any reference numbers):

This research is based on publicly available deidentified data and thus does not require an IRB approval.

## Data Availability

The raw data has been supplied as Supplemental Dataset Files.

## Supplemental Information

Supplemental information for this article can be found online at http://dx.doi.org/10.7717/peerj.4342#supplemental-information.

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
