# Peer review of "An observational study on risk of secondary cancers in chronic myeloid leukemia patients in the TKI era in the United States"

_PeerJ, doi:10.7717/peerj.4342_

## Round 0.1 · original submission · Major Revisions

Please address all the reviewers comments.

Reviewer 1 ·

Basic reporting

In this article ' A long term observational study on risk of secondary cancers in chronic myeloid leukemia patients in the TKI era in the United States' by Kumar et al., the authors used the updated SEER database, which released the chemotherapy treatment status of CML patients in April 2017. They systematically compared the SIR of secondary cancers status of these CML patients with the general population. They concluded that the CML patients have enhanced risk for SR, and indicated that TKI may not play a role in inducing the SRs. The paper clearly laid out the background information, their study design and nicely discussed the recent literatures in this field. Moreover, the limitations of this study in clearly stated in the discussion, which help better interpret the results and compare it with other studies.

The raw data shared is very clear, however, the two major figures of the paper are pretty small and blurry, which make it hard to read/comment on. The authors should improve the quality of these two figures.

One minor issue is the use of abreviations in the paper, i.e. SIR is not spelled out in its first appreance but the full name is spelled out more than once later on.

Experimental design

The main objective of this study is not clearly stated. As there are many previous studies that have already investigated the SR risks in CML patients, and some of these studies actually use the SEER database, the authors emphasize that one advantage of their analysis is that they included the chemotherapy treatment data for these patients. As these data are collected in the post TKI era, the authors stated their data could partially reflect whether TKI usage will affect SR development. However, the authors never state whether the aim of the study is to determine the impact of TKI on SR for CML patients. And if not, the authors should better explain what makes their study stand out compared to previous studies, i.e. what knowledge gap does this study fills in.

As the authors already mentioned in the discussion, their dataset only indicated whether patients were treated with chemotherapy or not. One does not know if the chemotherapy means TKI and should be caustious when making such assumptions, although TKI is one of the main chemo drugs used for CML in recent years. The authors should bring this explaination to introduction instead of mentioning them at the very end of discussion, to help readers understand the setup of the study from very begining.

If the authors would like to study the impact of TKI on SR development, a better way to analyze the data is to compare the SIR of SR rates between CML patients from pre-TKI era vs. those from post-TKI era, rather than comparing cancer patients data with the general population. This pre vs. post TKI comparion could eliminate factors such as the impact of genetic mutations due to CML on SR development.

Validity of the findings

The statistical analysis of this study is good in general. However, there are some issues that could be improved to better validate the findings.

1. In the patient selection criteria part, it is unclear why the authors exclude 141 patients since they developed 6 SRs. This has to be better explained.

2. It is not stated in the text how the authors get the SIR data of secondary cancer development in the general population.

3. The authors just compared the SIR of different groups but did not calculate the p-value of SIR to determine if the differences are statistically meaningful. The authors need to explain why p-value is not calculated.

Reviewer 2 ·

Basic reporting

I think the title uses the term “long-term observation study” but which was not well defined in the manuscript. One would also question if the follow-up duration for this group of patients with CML can be considered long enough to be called long term. I suggest the authors define the term accurately in the text and reconsider if this descriptive term is appropriate.

Experimental design

Cancers diagnosed in the period 2-12 months after the diagnosis of CML more likely could be a late presentation or remaining an overt characteristics of a synchronous double primary cancer. This period typically in an epidemiological study would be excluded from the calculation of secondary cancer. Therefore, I would like to see more likely "true" secondary cancer after the first follow-up year.

Line 116~117: The rationale to exclude this subgroup of patients seems not so justifiable.

After line 121: A subsection with a subheading of “Follow-up of Participants” or equivalent would be better to let readers understand the follow-up details.

Validity of the findings

Line 120 to 121: The authors stated that “A sensitivity analysis was conducted after excluding SCs which were diagnosed during the first year after diagnosis of CML to adjust for the surveillance bias.” Nevertheless, there are no ensuing results presented. I think it would be nice if a new figure 3 showing these results can be presented before acceptance.

Additional comments

The occurrence of chronic myeloid leukemia and chronic lymphocytic leukemia in the same patient is considered to be a rare event. However, this paper nicely quantitates the frequency of CLL after CML as 16/9200 = 0.17%, which, in my opinion, has added a contribution to the literature. Interestingly, case reports of CLL after CML also occurs in women, unlike the all-male result in this manuscript.

Table 1: Ethnicity: Would it be helpful to separate non-White into Black, Latin American and Asian-Pacific islanders?

Line 293 and line 265-266: In the Discussion, I think the authors can give us an idea of the estimated frequency of HSCT (which should be spelled out when first appears in the text) in patients with CML in the TKI era in the US.

Reviewer 3 ·

Basic reporting

Overall, the manuscript is very-well written with relevant introduction/references. It would be great if the authors could improve the quality of text in Fg. 1 & Fig. 2 as the current format is blurred/difficult to read.

Experimental design

The overall research question is very well defined and has clinical relevance.

Validity of the findings

The authors have extensively discussed the present study and compared with previous similar studies. The authors may want to add to the discussion as in what could potentially contribute to elevated risk for secondary cancers in CML in the TKI era.

Additional comments

Please see above.

---

## Round 0.2 · Minor Revisions

Please make Reviewer 2's suggested minor revisions.

Reviewer 1 ·

Basic reporting

The authors did a great job revising the manuscript. I have no further concerns and would recommend the publication of this work.

Experimental design

The authors did a great job revising the manuscript. I have no further concerns and would recommend the publication of this work.

Validity of the findings

The authors did a great job revising the manuscript. I have no further concerns and would recommend the publication of this work.

Additional comments

The authors did a great job revising the manuscript. I have no further concerns and would recommend the publication of this work.

Reviewer 2 ·

Basic reporting

This study, at least from the results in the presented Tables and Figures, does not investigate the impact of TKIs on the development of SCs. Please revise line 95, 96 to "This study aimed to analyze the development of SCs among patients with CML in the TKI era in the United States."

Experimental design

No comment

Validity of the findings

No comment

Additional comments

I think the Discussion Section has room to improve to make it more relevant and concise.

---

## Round 0.3 · accepted · Accept

Thank you for your effort to improve your paper according to the Editor and Reviewer suggestions